# Improvement of an Equivalent Circuit Model for Li-Ion Batteries Operating at Variable Discharge Conditions

**Gabriele Maria Lozito** [†], **Valentina Lucaferri** [*,†], **Francesco Riganti Fulginei** [†] **and Alessandro Salvini** [†]

Department of Engineering, University Roma Tre, Via Vito Volterra 62b, 00146 Rome, Italy; gabrielemaria.lozito@uniroma3.it (G.M.L.); francesco.rigantifulginei@uniroma3.it (F.R.F.); alessandro.salvini@uniroma3.it (A.S.)
* Correspondence: valentina.lucaferri@uniroma3.it
† These authors contributed equally to this work.

**Abstract:** A real time simulation of battery conditions is an essential step in the development of energy harvesting devices. Since it is not possible to have a direct measurement, the battery information, such as the remaining charge, need to be estimated by means of model-based estimation algorithms. Most of the existing models describing battery behaviour, are suitable only for a constant discharge current. This paper proposes a study of the dependence of the equivalent circuit model parameters on different discharge conditions. The model presented provides a powerful tool to represent the batteries' behaviour in energy harvesting systems, involving continuous charge and discharge cycles. The extraction of parameters was performed, starting from a set of reference curves generated in Matlab Simulink environment, referring to Li-ion technology batteries. The parameters were extracted by means of a cascade of global and local search identification algorithms. Finally, the relations describing parameters' behaviours as functions of the discharge current are presented.

**Keywords:** battery; circuital model; identification model

## 1. Introduction

In recent years, batteries have been widely used both for energy storage and as power sources in several applications, including electronic devices, electrical vehicles and renewable power systems. The main goals for each battery-powered system are to extend the battery's lifetime and include energy optimisation techniques in battery management systems (BMS). A deep overview of BMS applications is reported in [1], emphasising its importance in electric vehicles and the smart grids systems. The relevance of BMS in grid-connected battery-based system is stressed in [2] and a further strategy to control the battery charge and discharge based on optimisation techniques for energy accumulation in a stand-alone systems is proposed in [3].

All these applications require an accurate battery model to provide real-time information about the battery pack conditions; for instance, the state of charge (SoC), referring to the residual capacity of the battery, and the state of health (SoH), representing the ability of the battery to repeatedly provide its rated capacity over time. Since the internal characteristic and model parameters change according to the SoC and the operational temperature, the SoC estimation constitutes a fundamental step to understand how the battery works. Among the existing techniques, model-based methods are the most widely used to evaluate the SoC and study the dynamics of the battery. An exhaustive review of the recent techniques for estimating the model parameters and the SoC is described in [4]. Recently,

a robustness solution for representing the batteries' dynamic behaviour based on a combination of Kalman Filter algorithm and the equivalent circuit model was presented in [5], whereas for real time applications an accurate evaluation of SoC can be achieved by means of a novel estimator proposed in [6].

The estimate of battery SoC is a fundamental step to computing the terminal battery voltage. In this regard, many models have been developed in literature and the respective parameters are estimated by means of optimisation approaches, given that they are not measurable directly by sensors. In general, the existing models mainly fall into two categories: electrochemical and equivalent circuit ones. Electrochemical models are based on the chemical reactions that occur in a battery. Even though these models are useful to optimise the design of the battery and can achieve high precision, they are usually computationally intensive in both time and memory to solve partial differential equations [7,8]. On the other hand, the equivalent circuit models, (ECMs), are more straightforward and easier to implement than electrochemical ones. Moreover, they are especially suited to simulating the dynamic behaviour of the batteries by using resistance, capacitance, voltage source and other circuit components to form their circuits; more resistance capacitor (RC) networks usually improve the model accuracy [9]. Compared to electrochemical models that are characterised by partial differential equations, the ECMs are low order models; therefore, they result in better computational efficiency.

To identify the model parameters with low computational costs while preserving the accuracy, several identification procedures have been proposed in the literature, mainly involving non-linear least squares methods [10] and optimisation algorithms, including particle swarm optimisation [11] and genetic algorithms [12].

Most of these methods involve a set of parameters that are supposed to be constant for reducing the model complexity, but in practical applications they are subjected to change for different operating conditions, such as the state of charge (SoC—demonstrated in [13]); temperature; and ageing. Moreover, they present a limitation in the identification procedure, because they referred to constant current discharge curves and the parameters that are extracted are suitable only for fixed values of the discharge current.

To enhance the model accuracy, different sensitivity analyses have been proposed in the literature. In particular, the effects of changes in the model parameters on battery terminal voltage are analysed in [14]. Furthermore, a global sensitivity analysis of ECM parameters is studied in [8], reporting the strong dependence parameter on the SoC. This dependence is confirmed in [9], where a deep investigation of ECM parameters is presented, emphasising the importance of discharge current in voltage estimation.

In this work we analyse the parameters' dependencies on different load conditions, in order to make the ECM compliant with a variable discharge current. The proposed improvement allows one to identify the EMC model by starting from curves at constant current, readily available in the literature and easy to constructed by the knowledge of metadata. Our study follows two main steps. In the former, the identification of the parameters of ECM was performed for each reference voltage curve, referred to different values of discharge current and obtained in Simulink by using metadata. Then, a mathematical model expressing the parameters dependence on the discharge current values was built from these preliminary results by exploiting a trial and error approach. The proposed model was then validated by evaluating its error on battery voltage curves different from those used for the parameter extraction. The results show the possibilities of the adoption of the proposed model for the evaluation of battery dynamics in an energy harvesting systems.

The paper is structured as follows: in Section 2 the battery equivalent circuit model based on the state space equations is described; in Section 3 the identification process and the respective results are illustrated; in Section 4 the model proposed for the dependence of the circuital parameters on discharge current is presented and validated; finally, the conclusions are explained in Section 5.

## 2. Battery Equivalent Circuit Model

As stated before, several models have been developed for the battery according to the required accuracy and the application. Among these, the 2-RC networks model represents a good trade-off between the precision and the computational effort [15,16]. Figure 1 shows the ECM consisting of one C circuit on the left, used for SoC tracking, and two RC circuits on the right, used to simulate the I–V characteristics and transient response of the battery cell. The following subsections outline some of the battery's characteristics that are considered in the model.

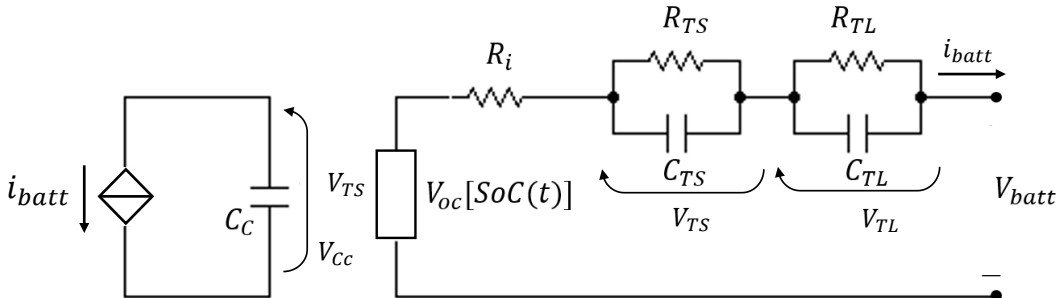

**Figure 1.** Equivalent circuit model: $C_c$ is the nominal capacity of the battery. The voltage across the capacitor, $V_{C_c}$, corresponds to the SoC of the battery cell. $V_{oc}$ is the open circuit voltage and it is a non linear function of the state of charge (SoC). The $R_i$, $R_{TS}C_{TS}$ and $R_{TL}C_{TL}$ are responsible for instantaneous voltage drop, short-time and long-time constants, respectively.

### 2.1. SoC Evaluation

The capacitance $C_{capacity}$ refers to the charge stored in the battery cell; the current source $i_{batt}$ represents the charge/discharge current of the battery cell; the voltage across the capacitor $V_{C_c}$ represents the SoC of the battery cell quantitatively and varies in the range of 0 V (i.e., the SoC is 0, standing for the battery being fully discharged) to 1 V (i.e., the SoC is 1, meaning that the battery is fully charged).

Therefore, the SoC can be calculated as:

$$C_{capacity}\frac{dSoC}{dt} = i_{batt}. \tag{1}$$

In this work, the $C_{capacity}$ corresponds to the usable capacity of the cell, and it is supposed to be a constant. This approximation, although it does not regard the non-linear capacity behaviours of the battery, such as the rate capacity and recovery effects, does not compromise the model accuracy, as will be demonstrated in the following sections. In order to include the non-linear capacity effects, a detailed extension of the EMC model is described in the Appendix A.

### 2.2. $V_{oc}$-SoC Characteristic

In general, the battery open-circuit voltage ($V_{oc}$) is a non-linear function of battery SoC and operating temperature. In this paper, the reference discharge curves are obtained by setting the temperature value as a constant. Therefore, only the dependence on the SoC is considered and the $V_{oc} = f[SoC(t)]$ is estimated by using:

$$V_{oc}[SoC(t)] = a_0 e^{-a_1 SoC} + a_2 SoC^3 + a_3 SoC^2 + a_4 SoC + a_5, \tag{2}$$

where the coefficients $a_0, a_1, a_2, a_3, a_4, a_5$ constitutes the parameter that need to be identified.

### 2.3. State Space Equations

The state space representation of the equivalent electrical model in Figure 1 is given by:

$$
\begin{bmatrix} \dot{V}_{TS} \\ \dot{V}_{TL} \\ \dot{SoC} \end{bmatrix} = \begin{bmatrix} -\dfrac{1}{R_{TS}C_{TS}} & 0 \\ 0 & -\dfrac{1}{R_{TL}C_{TL}} \\ 0 & 0 \end{bmatrix} \begin{bmatrix} V_{TS} \\ V_{TL} \end{bmatrix} + \begin{bmatrix} \dfrac{1}{C_{TS}} \\ \dfrac{1}{C_{TL}} \\ +\dfrac{1}{C_{Capacity}} \end{bmatrix} i_{batt} \tag{3}
$$

$$
V_{batt} = V_{oc}[SoC(t)] - R_i i_{batt} - V_{TS} - V_{TL}, \tag{4}
$$

where $V_{TS}$ and $V_{TL}$ are the voltages across the first and the second RC circuits, respectively; $C_{capacity}$ is the battery capacity; $i_{batt}$ is the current, which is assumed to be positive when the battery is discharging and negative during charging; and $V_{batt}$ is the terminal voltage.

## 3. Identification Process

The knowledge of unknown parameters $a_i$ with $i = 0...5$, $R_i$, $R_{TS}$, $C_{TS}$, $R_{TL}$ and $C_{TL}$, allows one to trace the discharge curve at constant current for a fixed operating condition. In order to extract the parameters, it is useful to manipulate the state space equations, providing the battery voltage $V_{batt}$ depending on time, $t$ and the 11 parameters $\theta = (a_0, a_1, a_2, a_3, a_4, a_5, R_i, R_{TS}, C_{TS}, R_{TL}$ and $C_{TL})$.

Therefore, the identification process consists in finding a combination of model parameters that minimises a cost function, represented by the squared error (SE) between the values of voltage $V_{batt}(t_n) = f(t_n, \theta)$, computed by means of Equation (4) for any time step $t_n$, and the reference/measured values of the battery voltage $V_n$:

$$
SE = \sum_{n=1}^{N} [V_n - f(t_n, \theta)]^2, \tag{5}
$$

where $N$ is the number of samples, $V_n$ is the vector of reference voltage values and $t_n$ is the vector time simulated samples. $f(t_n, \theta)$ is the computed voltage for each time step $t_n$.

The extraction of parameters from reference curves is a complex, multimodal problem, so in order to avoid local minima, a cascade of global and local search algorithms was implemented. In particular, the continuous flock optimisation algorithm, CFSO, is used for exploring the solutions space and reducing the space size.

It is a swarm intelligence-based strategy that is inspired by a very affirmed algorithm called the flock of starling optimisation (FSO). The advantage introduced by the CFSO algorithm, with respect to the FSO, is the availability of analytical expressions for the dynamic system poles. This provides the knowledge about the dynamic stability of the system, making it possible to have the swarm orbit around an area, converge in a pseudo-deterministic descent or escape from a local minimum, by simply tuning the stability parameters. The movement of particles is described by the following equations:

$$
v_k^j(t+1) = \omega v_k^j(t) + \lambda(p_{best}^j(t) - x_k^j(t)) + \gamma(g_{best}^j(t) - x_k^j(t)) + \sum_{m=1}^{N} h_{km} v_m^j(t) \tag{6}
$$

$$
x_k^j(t+1) = x_k^j(t) + v_k^j(t+1). \tag{7}
$$

By the index $j = 1...\Delta$, the dimension of the velocity position is taken into account. ($\Delta$ is the number of dimensions of the solution space). The index $k$ represents the particle of the N particles swarm. The symbols in the equations are from the standard convention used for the literature formulation of the PSO. The terms $\omega$, $\lambda$ and $\gamma$ represent, respectively, the inertial, cognitive and social coefficients. The three coefficients represent, respectively, the trend of a particle of maintaining its velocity, the attraction of a particle toward its personal best and the attraction of a particle toward the global best. The sum term at the far right is the weighted average of the velocities of a subset of particles belonging to the swarm. The continuous formulation of CFSO allows one to analyse the

system by means of domain transformations. A detailed description of the algorithm is reported in [17].

The main drawback of CFSO is the low degree of accuracy with respect deterministic algorithms; hence, the Levenberg Marquardt algorithm was implemented to obtain a better accuracy and find the optimal solution. The Levenberg–Marquardt, LM, algorithm was implemented to update the parameter vector, $\theta$, at each iteration step, $k$. The updating rule is:

$$\theta_{k+1} = \theta_k - (J_k^T J_k + \mu_k I)^{-1} J_k^T f_k, \tag{8}$$

where $J$ is the Jacobian matrix and $\mu$ is a coefficient regulating the the algorithm convergence. The advantages of this algorithm is the high convergence speed and strong stability.

*Data and Identification Method*

In the identification phase, we used the reference curves generated in MATLAB Simulink environment. In particular, by setting the metadata reported in Table 1, it is possible to extract the discharging curves, reported in Figure 2. Further details can be found in [18,19].

**Table 1.** Metadata used to extract the reference curves in Simulink.

| | | |
|---|---|---|
| Nominal Capacity [Ah] | 2.6 | 100 |
| Nominal Voltage [V] | 3.7 | 12 |
| Standard Discharge Current [A] | 1.27 | 43.5 |
| Maximum Discharge Current [A] | 2.6 | 500 |
| Operating temperature Discharge [°C] | $-20\ -60$ | $-15\ -50$ |
| Internal Impedance [mΩ] | $\leq 70$ | 8 |

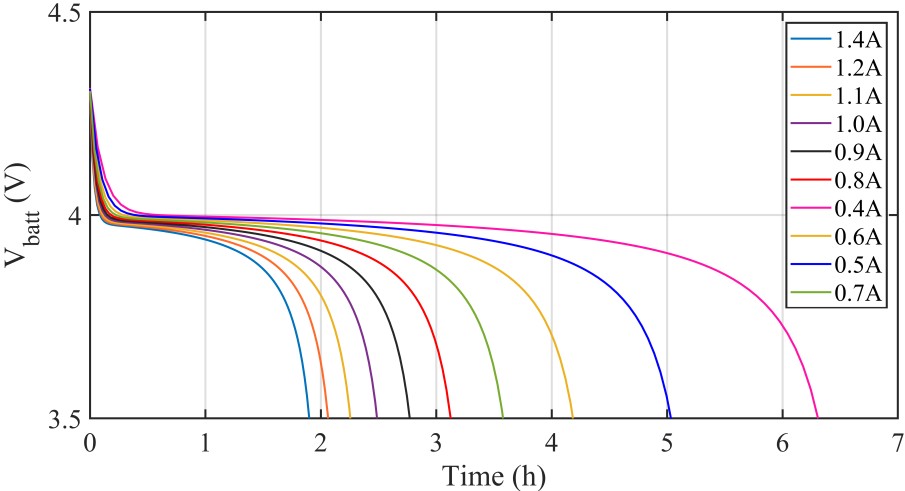

**Figure 2.** Discharge characteristics of Li-ion battery having a nominal voltage $V_n = 3.7$ V and a rated capacity $C_c = 2.6$ Ah for different values of discharge current.

As the behaviour of the proposed algorithm strongly depends on the guess values; it was run several times by setting a random guess values. An average of the solutions found was used for showing the trend of parameters. Results are reported in Table 2 for different values of discharge current.

**Table 2.** The 11 parameters found versus discharge current for a 2.6 Ah Li-ion battery.

| Parameters | | 0.5 A | 1.3 A | 1.9 A |
|---|---|---|---|---|
| $a_0$ | | $-6.350 \times 10^{-1}$ | $-7.990 \times 10^{-1}$ | $-9.620 \times 10^{-1}$ |
| $a_1$ | | $2.685 \times 10^{1}$ | $2.634 \times 10^{1}$ | $2.630 \times 10^{1}$ |
| $a_2$ | | $3.146 \times 10^{-1}$ | $5.160 \times 10^{-1}$ | $7.890 \times 10^{-1}$ |
| $a_3$ | | $-2.024 \times 10^{0}$ | $-2.597 \times 10^{0}$ | $-2.612 \times 10^{0}$ |
| $a_4$ | | $-1.269 \times 10^{0}$ | $-3.505 \times 10^{0}$ | $-3.419 \times 10^{0}$ |
| $a_5$ | | $7.205 \times 10^{0}$ | $9.898 \times 10^{0}$ | $9.518 \times 10^{0}$ |
| $R_i$ | $[\Omega]$ | $3.805 \times 10^{-4}$ | $3.768 \times 10^{-4}$ | $3.246 \times 10^{-4}$ |
| $R_{TS}$ | $[\Omega]$ | $7.810 \times 10^{-1}$ | $2.572 \times 10^{-1}$ | $1.543 \times 10^{-1}$ |
| $C_{TS}$ | $[F]$ | $4.616 \times 10^{2}$ | $4.523 \times 10^{2}$ | $4.476 \times 10^{2}$ |
| $R_{TL}$ | $[\Omega]$ | $1.551 \times 10^{1}$ | $1.326 \times 10^{1}$ | $1.068 \times 10^{1}$ |
| $C_{TL}$ | $[F]$ | $2.141 \times 10^{3}$ | $1.265 \times 10^{3}$ | $1.431 \times 10^{3}$ |

## 4. Model for Discharging Current Dependence

The parameters extracted were used to gain insight into their dependence on the discharge current. This allowed us to develop a model able to estimate the battery voltage for different discharge conditions. The procedure of modelling the trend of the battery voltage with current is performed as explained below:

- The first step deals with the observation of the behaviour of any of the parameters influenced by the discharge current: some known trends can be found.
- Therefore, for each of the parameters, a fitting procedure is performed for identifying a suitable polynomial or exponential function; the respective coefficients are extracted, too.
- Then, an error is calculated between reference curves, and samples are extracted through the identified closed-forms.
- Lastly, an analytical expression for the battery voltage is derived thanks to the knowledge of the trend of the parameters with current.

A regression analysis of some parameters for a Li-ion battery having 2.6 Ah is reported in Figure 3. The parameters $a_1$ and $R_i$ are considered constant, as can be seen from Table 1, while the others do not show any particular trend.

In general, the fitting functions assume one of the forms reported below; i.e., linear, quadratic or exponential:

$$f(x) = p_0 + p_1 x \tag{9}$$

$$g(x) = g_0 + g_1 x + g_2 x^2 \tag{10}$$

$$h(x) = a_0 e^{(-a_1 x)} + a_2. \tag{11}$$

In Table 3, the coefficients obtained by fitting the parameters trends and the respective coefficient of determination are illustrated.

**Table 3.** Coefficients obtained by fitting the parameters' trends and the respective coefficient of determination, $R^2$.

| | $a_0$ | $a2$ | $a_3$ | $R_{TS}$ | $C_{TS}$ | $R_{TL}$ |
|---|---|---|---|---|---|---|
| **Linear** | | | | | | |
| $p_0$ | $-5.527 \times 10^{-1}$ | | | | $4.612 \times 10^2$ | |
| $p_1$ | $-2.049 \times 10^{-1}$ | | | | $-7.032 \times 10^0$ | |
| **2 thorder polynomial** | | | | | | |
| $g_0$ | | $4.184 \times 10^{-1}$ | $-1.572 \times 10^0$ | | | $3.635 \times 10^1$ |
| $g_1$ | | $-2.340 \times 10^{-2}$ | $-1.233 \times 10^0$ | | | $-2.933 \times 10^1$ |
| $g_2$ | | $1.094 \times 10^{-1}$ | $3.584 \times 10^{-1}$ | | | $8.425 \times 10^0$ |
| **Exponential** | | | | | | |
| $a_0$ | | | | $1.319 \times 10^0$ | | |
| $a_1$ | | | | $2.056 \times 10^0$ | | |
| $a_2$ | | | | $1.399 \times 10^{-1}$ | | |
| $R^2$ | 0.9997 | 0.9557 | 0.8429 | 0.9986 | 0.9582 | 0.9396 |

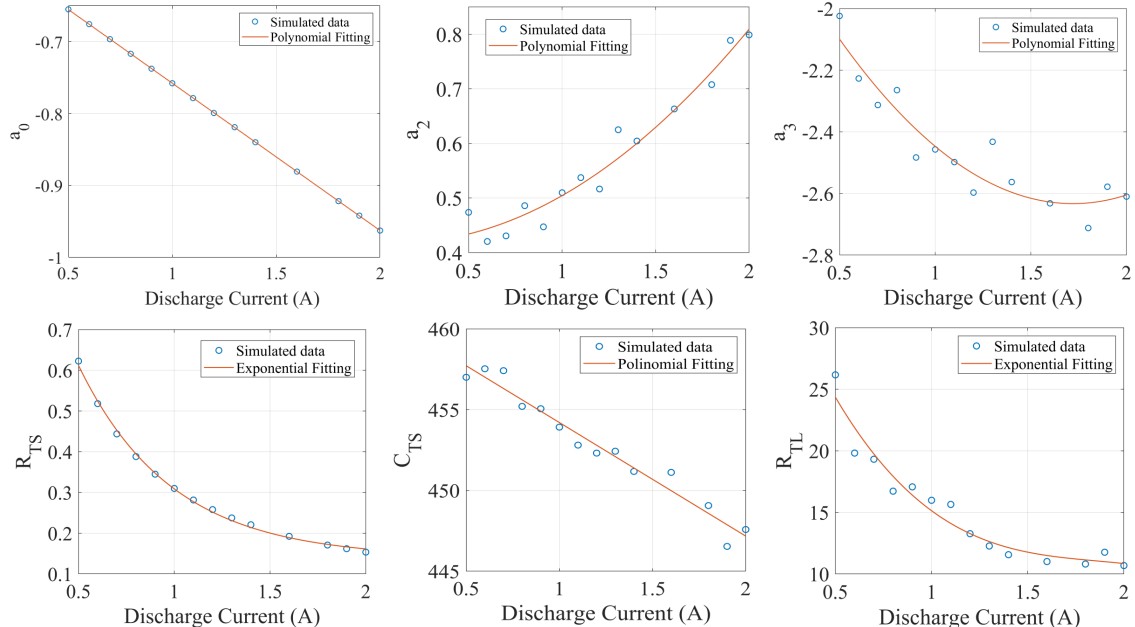

**Figure 3.** Parameters' trends for a Li-ion, C = 2.6 Ah battery: $a_0$, $a_2$, $a_3$, $C_{TS}$ and $R_{TL}$ are represented by a polynomial function: $f(x) = p_0 + p_1 x$, $g(x) = g_0 + g_1 x + g_2 x^2$; meanwhile, $R_{TS}$ is fitted by means of an exponential function: $h(x) = a_0 e^{-a_1 x} + a_2$.

## 4.1. Validation and Results

To validate the optimisation algorithm implemented and the proposed model, four cases of study were considered:

- Case 1: Comparison between optimisation algorithms. The CFSO was compared with the classical optimisation techniques, such as the genetic algorithm, GA, and the particle swarm optimisation, PSO.
- Case 2: Testing for different discharge currents. By considering the same Li-ion battery, C = 2.6 Ah, the model parameters were updated for different discharge currents, by applying the closed-form formula obtained in the previous section.
- Case 3: Trend parameters for Li-ion battery for electric vehicles. The identification process was implemented to extract the parameters trend of another battery technology, Li-ion C = 100 Ah. This confirms the trends achieved for Li-ion battery with C = 2.6 Ah.
- Case 4: Comparison between the proposed model and fixed parameters model. The discharge curves computed by our model and other models having fixed parameters but with different

discharge current values were compared for demonstrating the importance of updating the parameters.

### 4.1.1. Case 1: Comparison between Optimisation Algorithms

In order to demonstrate the good performance of CFSO, a comparative analysis against the PSO and GA for the initialisation of the search space to be investigated by the LM is reported. Both the PSO and the GA were used considering a set of 50 agents (i.e., 50 particles for the PSO and 50 individuals for the GA). All approaches ran 250 iterations: 150 for each global optimisation algorithm (CFSO, PSO and GA) and 100 for the refinements via LM. Results are summarised in Table 4.

**Table 4.** Comparison against classic global optimisation algorithms.

|                       | CFSO + LM | PSO + LM | GA + LM |
|-----------------------|-----------|----------|---------|
| Iterations            | 250       | 250      | 250     |
| RMSE                  | 0.0447    | 0.0801   | 0.0503  |
| Computational time [s]| 1.45      | 1.21     | 2.69    |

### 4.1.2. Case 2: Test for Different Discharge Currents

During the validation phase, we considered four curves referring to discharge currents of 0.2 , 0.3, 1.5 and 1.7 A that were not involved in the identification step. As can be seen from Figures 4 and 5 the error was always below $1 \times 10^{-2}$. The results related to the test curves are reported in Table 5, where the Mean Squared Error, MSE, is very low—about $1 \times 10^{-2}$.

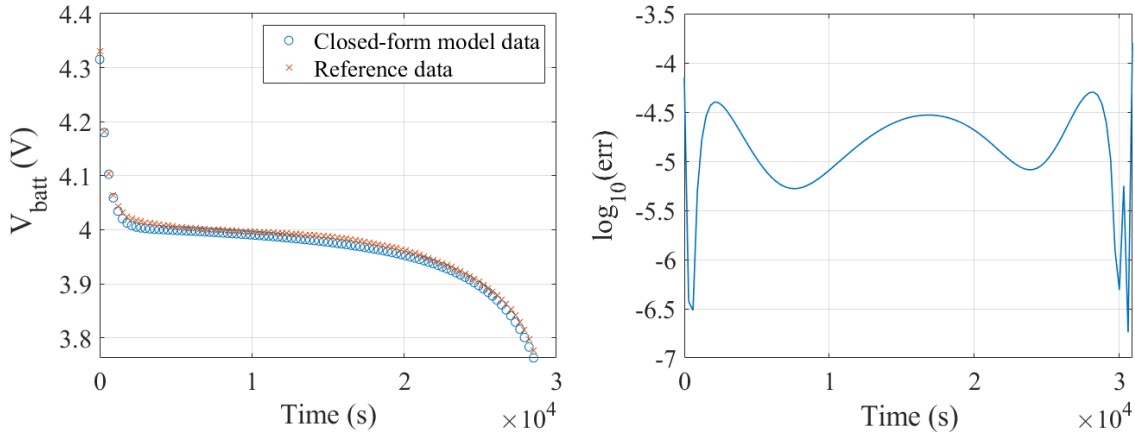

**Figure 4.** Test for a discharge current of 0.3 A: comparison between the voltage estimated by means of a closed-form parameter-based model and the voltage reference values (on the **left**). Absolute Error in logarithmic scale between simulated and reference values (on the **right**).

**Table 5.** MSEs for different cases of study.

| Discharge Current [A] | MSE |
|-----------------------|-----|
| 0.2 | $6.2 \times 10^{-3}$ |
| 0.3 | $8.3 \times 10^{-3}$ |
| 1.5 | $2.1 \times 10^{-2}$ |
| 1.7 | $3.3 \times 10^{-2}$ |

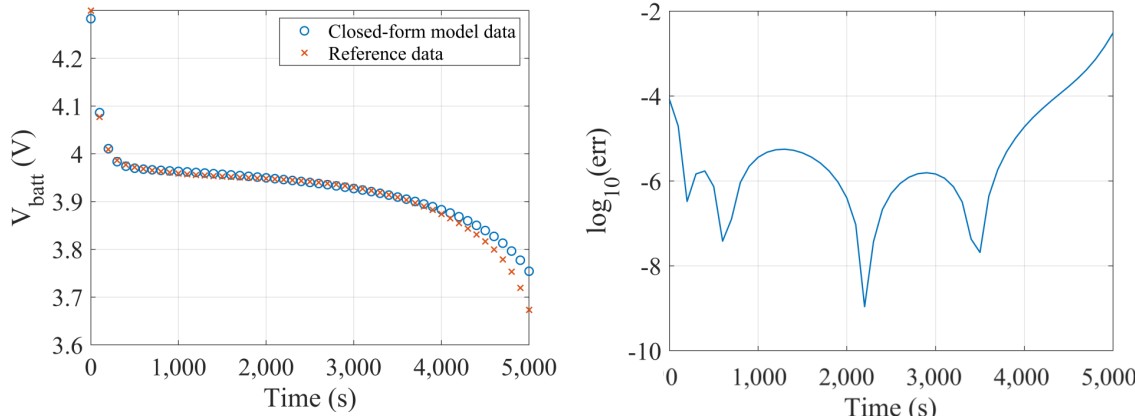

**Figure 5.** Test for a discharge current of 1.7 A: comparison between the voltage estimated by means of a closed-form parameter-based model and the voltage reference values (on the **left**). Absolute error in logarithmic scale between simulated and reference values (on the **right**).

### 4.1.3. Case 3: Trend Parameters for Li-Ion Battery for Electric Vehicles

A further validation step is considered by referring to a battery technology different from that one studied in previous section: Li-ion C = 100 Ah for electric vehicles applications. Figure 6 shows the parameters' trends and demonstrates that they are similar to the ones obtained from the first applications. For sake of brevity, we reported the results for some parameters only.

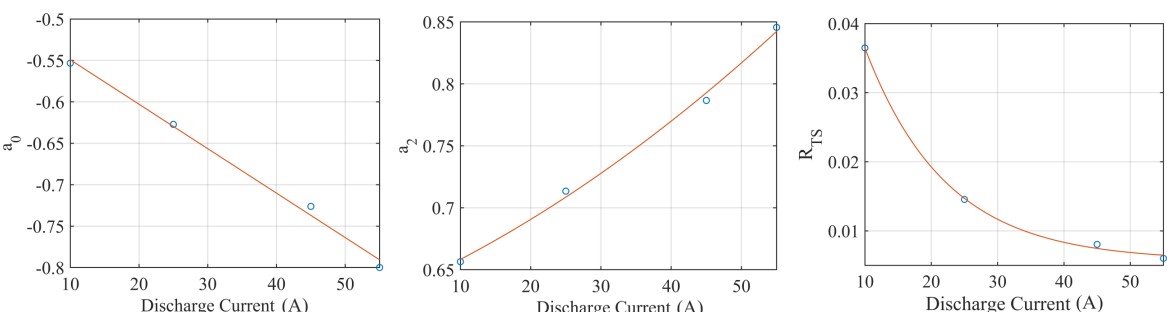

**Figure 6.** Parameters' trends for Li-ion, C = 100 Ah: $a_0$ and $a_2$ are represented by a polynomial; $R_{TS}$ is fitted by means of an exponential function.

### 4.1.4. Case 4: Comparison between the Proposed Model and Fixed Parameters Model

In order to demonstrate the good performance of the model proposed here, a comparison between the our model implementing the parameters variation at different current values (10 and 25 Ampere) and the model with a fixed parameters values is reported in Figure 7.

The results show that the update of parameters, when the discharge current changes, is necessary. The model maintaining a fixed parameter is not able to approximate correctly the discharge curve.

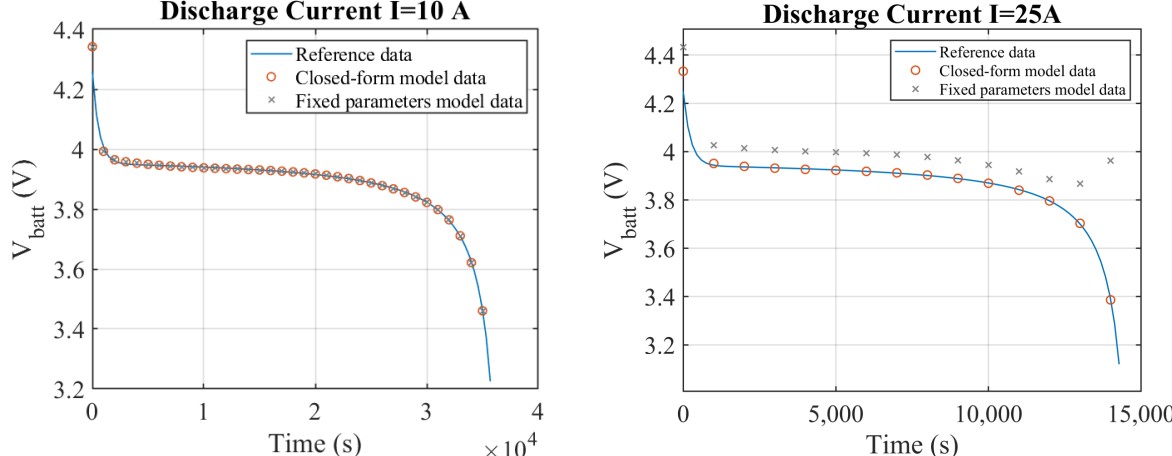

**Figure 7.** Discharge curves for two different current values: comparison between the results obtained through the proposed closed-form model and the fixed parameters model.

## 5. Conclusions

This paper has pursued the improvement of an equivalent circuit model able to predict the behaviour of battery voltage with varying discharge currents. An efficient identification procedure based on an optimisation algorithm has been developed and applied to a set of reference curves easily generated in the Simulink environment, starting with the knowledge of metadata; a combination of the CFSO and the Levenberg–Marquardt algorithm allowed us to extract the value of each of the parameters influencing the battery voltage at different discharge currents. Therefore, the trends observed have been used to derive a mathematical expressions able to furnish an analytical solution not only for each of the parameters, but in general, for the voltage of any battery. This makes the ECM proposed in literature valid for any applications involving a variable terminal current. In particular, such improvements are important in battery energy storage systems, where a battery needs to be interfaced with a DC/DC converter, whose current varies continuously in order to allow the charge and discharge. The drawback of this method is that there is no physical meaning in the relationship obtained between parameters and the discharge current.

**Author Contributions:** Conceptualisation, V.L. and G.M.L.; methodology, V.L.; software, F.R.F.; validation, V.L.; formal analysis, V.L.; investigation, G.M.L.; resources, V.L.; data curation, G.M.L.; writing–original draft preparation, V.L.; writing–review and editing, G.M.L. and F.R.F.; supervision, F.R.F. and A.S. All authors have read and agreed to the published version of the manuscript.

**Funding:** This research received no external funding.

**Conflicts of Interest:** The authors declare no conflict of interest.

## Appendix A

The equivalent circuit model can be easily extended in order to consider a variable capacity. It can be integrated by inserting the Kinetic Battery model, which takes into account the non linear phenomena occurring in a battery, the rate and recovery effects.

As shown in Figure A1, the battery model is assumed to be composed of two charge wells, where the charge is distributed with a capacity ratio, $c$ ($0 < c < 1$): one contains the available charge and supplies the load directly, while the other contains the bound charge and refills the available charge well (recovery effect) through a valve, $k$. The charge exchanged between the wells depends on the difference in their heights, $\delta = h_1 - h_2$ and $k$. $h_1$ represents the state of charge, SoC, and when it becomes zero, the battery is full discharged.

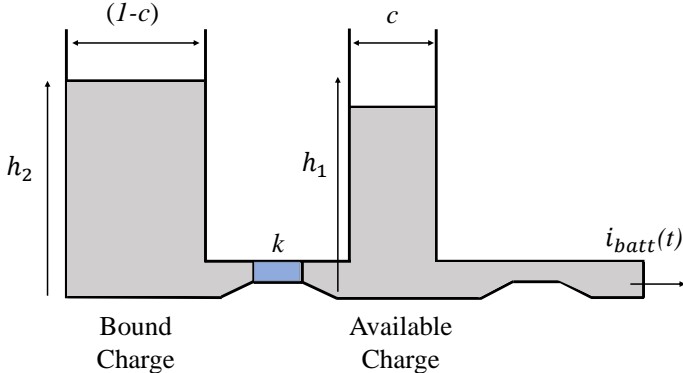

**Figure A1.** Kinetic battery model: the battery is represented as structure composed by two charge wells: bound charge and available charge. The capacity ratio, $c$, defines the charge distribution.

The change of charge between the wells is expressed as follows:

$$\begin{cases} \frac{\partial x_1}{\partial t} = -i(t) + k[h_2(t) - h_1(t)] \\ \frac{\partial x_2}{\partial t} = -k[h_2(t) - h_1(t)], \end{cases} \tag{A1}$$

where $x_1$ and $x_2$ are the total charge in available and bound charge well, respectively, from which comes $h_1 = \frac{x_1}{c}$ and $h_2 = \frac{x_2}{(1-c)}$. When the battery is discharged with a current, $i_{batt}(t)$, the available charge, decreases; thus, the SoC and voltage reduce (rate effect). When the load is removed, the charge flows from the bound charge well to the available charge well until $h_1$ and $h_2$ are equal; this is the recovery effect. By assuming the following initial conditions $x_{1,0} = x_1(t=0) = c \cdot C$ $x_{2,0} = x_2(t=0) = (1-c) \cdot C$ and $x_0 = x_{1,0} + x_{2,0}$ with C representing the total battery capacity, the differential equations can be solved for a constant discharge current, $I$, for a period $t_0 < t < t_1$. The discharge completes when $x_1$ becomes zero. The unavailable charge of the battery can be expressed as:

$$\begin{cases} C_{un} = (1-c) \cdot \delta(t) \\ \delta(t) = h_2(t) - h_1(t) = \frac{x_2}{(1-c)} - \frac{x_1}{c}. \end{cases} \tag{A2}$$

By solving equations and measuring the load current $i_{batt}(t)$, the state of charge of the battery can be computed by using:

$$SOC(t) = \frac{C_{available}(t)}{C_{max}} = SOC_{in}(t) - \frac{1}{C_{max}} \left[ \int i(t)dt + C_{un}(t) \right], \tag{A3}$$

where $C_{available}$ and $C_{max}$ are the available and maximum battery capabilities. The SoC decreases when it delivers charge to load, which is expressed by current integration term.

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
