# Peer review of "Improvement of an Equivalent Circuit Model for Li-Ion Batteries Operating at Variable Discharge Conditions"

_electronics, doi:10.3390/electronics9010078_

Round 1

Reviewer 1 Report

N/A

Reviewer 2 Report

The comments have been successfully addressed in the revised manuscript. Suggest for publication in current format.

Reviewer 3 Report

In this manuscript, it is analysed the parameters dependence on different load conditions in Li-Ion batteries, in order to make the electric circuit model compliant with a variable discharge current. An efficient identification procedure based on an optimization algorithm has been developed and applied to a set of reference curves easily generated in Simulink environment starting by the knowledge of metadata; a combination of the continuos of flock optimization algorithm and the Levenberg-Marquardt algorithm allowed to extract the value of each of the parameters influencing the battery voltage at different discharge currents. The manuscript is well written. The Introduction and Conclusion sections are clear and concise. The manuscript is publishable in Electronics.

This manuscript is a resubmission of an earlier submission. The following is a list of the peer review reports and author responses from that submission.

Round 1

Reviewer 1 Report

The manuscript described a study of improvement of an Equivalent circuit model for LIBs at different discharge rates through an optimization algorithm. The simulation results fit well for the used reference curves. However, whether the obtained parameters will be suitable for other batteries is a question mark. For example, in the experiment, a 2.6 Ah LIB and different discharge current varying from 0.4 A~1.4 A was applied, corresponding to C/2 ~C/6.5 rate. By looking at the data in Fig 2, it can be seen that the cell capacity didn’t decrease when the rate increased from C/6.5 to C/2, which means that the cell used in this experiment could be a power cell. In other way, if an energy cell was used in the experiment, a capacity drop will happen when it is experiencing a rate change from C/6.5 to C/2. So the main question is that:  the experiment is designed to validate the simulation parameters at different discharge rate, the cell used here is not that sensitive to the rate change.

Reviewer 2 Report

1.The authors discussed that 2-RC network represents a good trade-off between the precision and the computational effort. The authors are suggested to cite proper reference to support this conclusion.

2.For Figure 1, the authors should explain the meaning of elements, such as RTS, RTL, CTS and CTL, in the circuit.

3.Why did the authors choose the capacity of 2.6 Ah for the study?

4.For Figure 3, what are R2 values for the fitting functions? 

5.For validation process, in addition to higher discharge currents (>1.4A) of 1.5A and 1.7A, the authors should also check lower currents (<0.4A), such as 0.3A and 0.2A.
